# Predicting resistance to fluoroquinolones among patients with rifampicin-resistant tuberculosis using machine learning methods

Shiying You[1,2], Melanie H. Chitwood[2,3], Kenneth S. Gunasekera[2,3], Valeriu Crudu[4], Alexandru Codreanu[4], Nelly Ciobanu[4], Jennifer Furin[5,6], Ted Cohen[2,3], Joshua L. Warren[2,7], Reza Yaesoubi[1,2]*

1 Department of Health Policy and Management, Yale School of Public Health, New Haven, Connecticut, United States of America, 2 Public Health Modeling Unit, Yale School of Public Health, New Haven, Connecticut, United States of America, 3 Department of Epidemiology of Microbial Diseases, Yale School of Public Health, New Haven, Connecticut, United States of America, 4 Phthisiopneumology Institute, Chisinau, Republic of Moldova, 5 Department of Medicine, Case Western Reserve University, Cleveland, Ohio, United States of America, 6 Department of Global Health and Social Medicine, Harvard Medical School, Boston, Massachusetts, United States of America, 7 Department of Biostatistics, Yale School of Public Health, New Haven, Connecticut, United States of America

* reza.yaesoubi@yale.edu

**Data Availability Statement:** The dataset used in this study is provided in the supplementary file S1

## Abstract

### Background

Limited access to drug-susceptibility tests (DSTs) and delays in receiving DST results are challenges for timely and appropriate treatment of multi-drug resistant tuberculosis (TB) in many low-resource settings. We investigated whether data collected as part of routine, national TB surveillance could be used to develop predictive models to identify additional resistance to fluoroquinolones (FLQs), a critical second-line class of anti-TB agents, at the time of diagnosis with rifampin-resistant TB.

### Methods and findings

We assessed three machine learning-based models (logistic regression, neural network, and random forest) using information from 540 patients with rifampicin-resistant TB, diagnosed using Xpert MTB/RIF and notified in the Republic of Moldova between January 2018 and December 2019. The models were trained to predict the resistance to FLQs based on demographic and TB clinical information of patients and the estimated district-level prevalence of resistance to FLQs. We compared these models based on the optimism-corrected area under the receiver operating characteristic curve (OC-AUC-ROC). The OC-AUC-ROC of all models were statistically greater than 0.5. The neural network model, which utilizes twelve features, performed best and had an estimated OC-AUC-ROC of 0.87 (0.83,0.91), which suggests reasonable discriminatory power. A limitation of our study is that our models are based only on data from the Republic of Moldova and since not externally validated, the generalizability of these models to other populations remains unknown.

Dataset.xlsx. All other information needed to replicate the findings of this study is provided in the main text and S1 Text.pdf.

**Funding:** This study was supported by the United States Agency for International Development (www.usaid.gov) through the TREAT TB Cooperative Agreement No. GHN-A-00-08-00004 to TC and JW. RY was supported by K01AI119603 and TC by R01AI112438 and R01AI146555, all from the National Institute of Allergy and Infectious Diseases (www.niaid.nih.gov). KSG was supported by the US National Institutes of Health through the Eunice Kennedy Shriver National Institute of Child Health and Human Development (www.nichd.nih.gov) [F30HD105440] as well as the Medical Scientist Training Program [T32GM007205] through the National Institute of General Medical Sciences (www.nigms.nih.gov). The funders had no role in study design, data collection and analysis, decision to publish, or preparation of the manuscript.

**Competing interests:** I have read the journal's policy and the authors of this manuscript have the following competing interests: VD, AC, NC are employed by Phthisiopneumology Institute.

## Conclusions

Models trained on data from phenotypic surveillance of drug-resistant TB can predict resistance to FLQs based on patient characteristics at the time of diagnosis with rifampin-resistant TB using Xpert MTB/RIF, and information about the local prevalence of resistance to FLQs. These models may be useful for informing the selection of antibiotics while awaiting results of DSTs.

## Introduction

Tuberculosis (TB), an infectious disease caused by *M. tuberculosis* bacterium, is one of the top ten leading causes of death worldwide [1]. Despite recent declines in global TB incidence, drug-resistant TB continues to pose major challenges to TB control in several countries [1–3]. An estimated 3.3% of incident cases with no previous treatment and 18% of incident cases with previous treatment had multi-drug resistant or rifampicin-resistant TB (MDR/RR-TB) in 2019 [1]. The treatment of drug-resistant TB is challenging (with the success rate of 57% for MDR/RR-TB in 2019) and requires long courses (between 9–20 months) of regimens consisting of multiple antibiotics [1,4]. Many of these second-line antibiotics are associated with severe side effects [2,3].

The advent and adoption of a molecular test for the rapid detection of TB and the resistance to rifampicin (Xpert MTB/RIF) has enabled TB programs to detect individuals with RR-TB at the time of TB diagnosis [1]. The rapid detection of resistance to other anti-TB agents critical for the selection of effective treatment for individuals with RR-TB, remains a serious challenge. Phenotypic drug-susceptibility tests (DSTs) are not commonly available in many low-resource settings [1,5]. Even when available, culture-based methods take up to 12 weeks to provide results and may only be pursued routinely among a subset of culture-positive patients [6,7]. Unlike phenotypic DSTs, genotypic DSTs, such as line probe assays, can provide results within hours but these tests are also not available in many settings [4,5,8].

Given the limitations of current DSTs, in most settings, the treatment of individuals detected by Xpert MTB/RIF as having RR-TB remains empiric (i.e., without the knowledge about their full drug-susceptibility profile) and according to standardized regimens [9]. These standardized regimens are often determined at the global level, with recommendations that the local epidemiology of drug-resistant TB, which is often unknown, be used to determine final composition [9,10]. The prevalence of resistance to second-line anti-TB agents varies markedly across different counties and regions [10–17]. For example, in a population-based study conducted in Azerbaijan, Bangladesh, Belarus, Pakistan, and South Africa, the prevalence of resistance to pyrazinamide and fluoroquinolones (two of the antibiotics included in the standardized shorter regimen recommended by the World Health Organization (WHO) for people newly diagnosed with RR-TB) [9] varied substantially between settings: 2.1–3.0% for pyrazinamide, 1.0–16.6% for ofloxacin, 0.5–12.4% for levofloxacin, and 0.9–14.6% for moxifloxacin [16]. In communities where the prevalence of resistance to the antibiotics included in the standardized second-line regimens is high, the use of standardized regimens results in many individuals with RR-TB receiving antibiotics that do not match their drug-susceptibility profile [7,12]. These patients are at higher risk of mortality and may experience a longer duration of infectiousness, which could lead to further transmission of drug-resistant TB [18–23]. Receiving inappropriate treatment would also increase the risk of functional monotherapy and of selection for additional drug resistance [12,24–27].

One potential approach for improving the selection of antibiotics for patients with RR-TB is to customize the treatment regimen based on observable characteristics of patients that are associated with drug susceptibility profiles [7,10,12]. Prior studies have identified patient characteristics that are associated with an increased or decreased risk of resistance including age [28–33], sex [29,31,32,34–36], education [29,34], rural/urban residence [30,34,37,38], geographic location [39,40], occupation/employment status [29,34,38], living condition (e.g., living in a household with only one room) [28,30], smoking [29,34,37], history of detention [30,32], infection with HIV [28,30,32–34], previous anti-TB treatment, history of anti-TB treatment failure, and previous hospitalization [41,42]. Yet, evidence is limited on whether these risk factors can be used in practice, at the point-of-care, to identify patients with TB that is likely resistant to specific second-line anti-TB agents and to inform individualized treatment recommendations [43].

In this study, we examine whether data from national phenotypic surveillance systems of drug-resistant TB can be used to develop predictive models for identifying resistance to fluoroquinolones (a critical second-line class of anti-TB agents) among patients diagnosed with RR-TB using Xpert MTB/RIF. We develop and evaluate these predictive models using prospectively-collected data on the demographic and health status of 2,518 patients with culture-positive TB notified between January 2018 and December 2019 in the Republic of Moldova.

To evaluate whether these predictive models could improve the selection of antibiotics, we consider a scenario where a fluoroquinolone (FLQ) would be replaced with delamanid (DLM) in the empiric RR-TB treatment regimen if additional resistance to FLQs is suspected (this follows the hierarchy of the WHO grouping of anti-TB agents) [9]. We measure the utility of these predictive models based on their ability to 1) increase the proportion of patients with RR-TB who receive an appropriate treatment regimen (i.e., a regimen that matches the susceptibility of their *M. tuberculosis* strain to FLQs) and 2) reduce the use of DLM to minimize selective pressure for DLM resistance.

## Methods

### Data source and study population

Our data were collected in the course of routine activities of the national tuberculosis program in the Republic of Moldova. TB incidence in the Republic of Moldova was estimated at 80 cases per 100,000 population in 2019, one of the highest rates in the European Region [1, 44]. Moldova also has one of the highest incidence rates of MDR/RR-TB in the world. In 2019, 33% of new cases and 60% of retreatment cases in Moldova had MDR/RR-TB [1]. Substantial efforts have been made in the country to combat the rise of MDR/RR-TB including the widespread adoption of Xpert MTB/RIF for rapid TB detection and universal use of phenotypic drug susceptibility testing for all patients with culture-positive TB [45].

Our dataset includes the records of 2,518 individuals with incident culture-positive TB detected between January 2018 and December 2019 (Table 1 and S1 Dataset). The record of each individual includes demographic information (e.g., age, sex, education, occupation, incarceration), diagnostic test results (e.g., results of Xpert MTB/RIF, microscopy), and additional clinical descriptors (e.g., location and severity of infection). To develop and evaluate the predictive models proposed here, we only included individuals with RR-TB diagnosed using Xpert MTB/RIF who had conclusive DST results to determine resistance or susceptibility to FLQs (Fig 1).

### Ethics statement

Individuals being evaluated for suspected pulmonary TB during the time period of our study were approached for enrollment by physicians and nurses trained in informed consent. Written consent was provided to allow access to routinely collected basic demographic, residential,

**Table 1. Demographic information, TB-related information, and test results in the data collected from the national tuberculosis surveillance system in the Republic of Moldova between January 2018 to December 2019.**

| Variable | Individuals with RR-TB and confirmed DST for FLQs, N = 540) | | Individuals with RR-TB and confirmed FLQ-resistance (N = 101) | |
|---|---|---|---|---|
| | Mean / Freq | SD / % | Mean / Freq | SD / % |
| **Demographics** | | | | |
| Age | 42.81 | 12.44 | 43.25 | 11.83 |
| Sex | | | | |
| Male | 424 | 78.52% | 72 | 71.29% |
| Female | 116 | 21.48% | 29 | 28.71% |
| Occupation | | | | |
| Employed | 72 | 13.33% | 11 | 10.89% |
| Disabled | 47 | 8.7% | 11 | 10.89% |
| Retired | 41 | 7.59% | 11 | 10.89% |
| Student | 9 | 1.67% | 3 | 2.97% |
| Unemployed | 370 | 68.52% | 64 | 63.37% |
| Missing | 1 | 0.19% | 1 | 0.99% |
| Number of household contacts | | | | |
| 0 | 114 | 21.11% | 27 | 26.73% |
| 1 | 126 | 23.33% | 24 | 23.76% |
| 2 | 100 | 18.52% | 17 | 16.83% |
| 3 | 71 | 13.15% | 8 | 7.92% |
| 4 | 43 | 7.96% | 10 | 9.90% |
| 5+ | 52 | 9.63% | 9 | 8.91% |
| Missing | 34 | 6.3% | 6 | 5.94% |
| Number of household contacts 18 or younger | | | | |
| 0 | 298 | 55.19% | 60 | 59.41% |
| 1 | 59 | 10.93% | 8 | 7.92% |
| 2 | 56 | 10.37% | 9 | 8.91% |
| 3 | 19 | 3.52% | 6 | 5.94% |
| 4+ | 12 | 2.22% | 3 | 2.97% |
| Missing | 96 | 17.78% | 15 | 14.85% |
| Education | | | | |
| Primary | 176 | 32.59% | 31 | 30.69% |
| Secondary | 235 | 43.52% | 47 | 46.53% |
| Specialized secondary | 96 | 17.78% | 16 | 15.84% |
| Higher education | 18 | 3.33% | 4 | 3.96% |
| No education | 10 | 1.85% | 2 | 1.98% |
| Missing | 5 | 0.93% | 1 | 0.99% |
| Satisfactory living condition | | | | |
| Yes | 263 | 48.70% | 48 | 47.52% |
| No | 208 | 38.52% | 35 | 34.65% |
| Missing | 69 | 12.78% | 18 | 17.82% |
| Outside Moldova for more than 3 months | | | | |
| Yes | 71 | 13.15% | 15 | 14.85% |
| No | 444 | 82.22% | 82 | 81.19% |
| Missing | 25 | 4.63% | 4 | 3.96% |
| Residing in urban area | | | | |
| Yes | 244 | 45.19% | 45 | 44.55% |
| No | 295 | 54.63% | 56 | 55.45% |

(*Continued*)

**Table 1.** (Continued)

| Variable | Individuals with RR-TB and confirmed DST for FLQs, *N* = 540) | | Individuals with RR-TB and confirmed FLQ-resistance (*N* = 101) | |
|---|---|---|---|---|
| | Mean / Freq | SD / % | Mean / Freq | SD / % |
| Missing | 1 | 0.19% | - | - |
| Homeless | | | | |
| Yes | 62 | 11.48% | 13 | 12.87% |
| No | 466 | 86.30% | 87 | 86.14% |
| Missing | 12 | 2.22% | 1 | 0.99% |
| Receiving money assistance | | | | |
| Yes | 151 | 27.96% | 32 | 31.68% |
| No | 341 | 63.15% | 62 | 61.39% |
| Missing | 48 | 8.89% | 7 | 6.93% |
| Previously incarcerated | | | | |
| Yes | 80 | 14.81% | 12 | 11.88% |
| No | 413 | 76.48% | 80 | 79.21% |
| Missing | 47 | 8.7% | 9 | 8.91% |
| Residing in a district with low, median, or high prevalence of resistance to FLQs[1] | | | | |
| Low (<10%) | 105 | 19.44% | 4 | 3.96% |
| Medium (10%-20%) | 88 | 16.30% | 14 | 13.86% |
| High (>20%) | 292 | 54.07% | 80 | 79.21% |
| Missing | 55 | 10.19% | 3 | 2.97% |
| **TB-related information** | | | | |
| TB location | | | | |
| Pulmonary | 527 | 97.59% | 99 | 98.02% |
| Extra-pulmonary | 5 | 0.93% | 1 | 0.99% |
| Missing | 8 | 1.48% | 1 | 0.99% |
| TB type | | | | |
| New case | 340 | 62.96% | 61 | 60.40% |
| Relapse case | 138 | 25.56% | 17 | 16.83% |
| Return after default | 48 | 8.89% | 14 | 13.86% |
| Treatment failure | 10 | 1.85% | 7 | 6.93% |
| Initiated treatment abroad | 4 | 0.74% | 2 | 1.98% |
| **Test Results** | | | | |
| Microscopy | | | | |
| Positive | 250 | 46.30% | 42 | 41.58% |
| Negative | 234 | 43.33% | 49 | 48.51% |
| Missing | 56 | 10.37% | 10 | 9.91% |
| Xpert | | | | |
| Positive | 540 | 100% | 101 | 100% |
| Negative | - | - | - | - |
| Missing | - | - | - | - |
| Xpert-RIF[2] | | | | |
| Positive | 540 | 100% | 101 | 100% |
| Negative | - | - | - | - |
| Missing | - | - | - | - |
| Rifampicin resistance detected by culture[3] | | | | |
| Positive | 464 | 85.93% | 83 | 82.18% |
| Negative | 27 | 5.00% | - | - |

(*Continued*)

**Table 1.** (Continued)

| Variable | Individuals with RR-TB and confirmed DST for FLQs, N = 540) | | Individuals with RR-TB and confirmed FLQ-resistance (N = 101) | |
|---|---|---|---|---|
| | Mean / Freq | SD / % | Mean / Freq | SD / % |
| Missing | 49 | 9.07% | 18 | 17.82% |
| FLQ resistance[4] | | | | |
| Positive | 101 | 18.7% | 101 | 100% |
| Negative | 439 | 81.3% | - | - |
| Missing | - | - | - | - |

[1]Prevalence of resistance to FLQs is calculated using the data between January 2018 and December 2019. The 'missing' category represents individuals with no information about their district of residence or who live in districts with fewer than 5 notified RR-TB during this period.

[2]All patients received Xpert MTB/RIF as diagnostic test, which also reveals RIF susceptibility profile during diagnosis. However, some Xpert negative patients may also prove culture positive and later be detected through DST as rifampicin-resistant.

[3]Resistance to rifampicin was assumed if either or both LJ and MGIT culture tests were positive (see §S1.1 in S1 Text for additional details). We note that given that these culture tests have imperfect sensitivity and specificity, it is possible that a small number of individuals who are diagnosed with rifampicin-resistant TB through Xpert-MTB/RIF have a negative culture test.

[4]Resistance to FLQs was assumed if resistance to at least one of the FLQs (i.e., ofloxacin, levofloxacin, and/or moxifloxacin) was detected (see §S1.1 in S1 Text for additional details).

and epidemiological data. This study was approved by the Ethics Committee of Research of the Phthisiopneumology Institute in Moldova and the Yale University Human Investigation Committee (No. 2000023071).

## Predictors and outcomes

Our goal was to examine whether resistance to FLQs can be predicted for an individual diagnosed with RR-TB using Xpert MTB/RIF based on observable characteristics of that individual

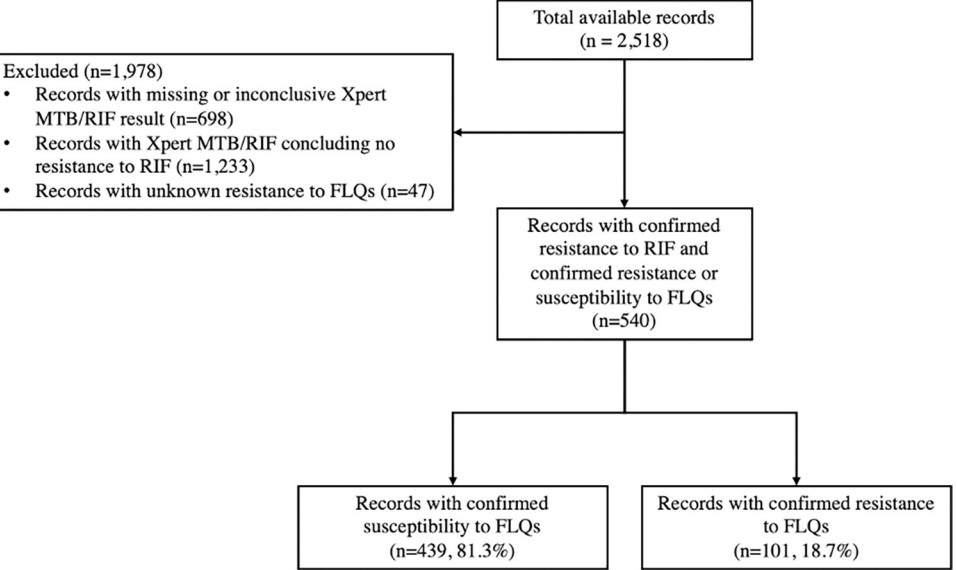

**Fig 1. Flowchart of inclusion criteria.** RIF: rifampicin, FLQ: a fluoroquinolone (ofloxacin, levofloxacin, or moxifloxacin). Resistance/susceptibility to RIF was determined based on the results of Xpert MTB/RIF test. Resistance/susceptibility to FLQs was determined based on the results of LJ and/or MGIT culture tests for ofloxacin, levofloxacin, and moxifloxacin. Resistance to FLQs was assumed if the resistance to least one of these three drugs was detected (see §S1.1 in S1 Text for additional details).

(commonly referred to as *features*). To develop our predictive models, we considered the following features, which are observable at the point-of-care when the selection of antibiotics for the empiric treatment is determined:

1. Demographic information, including age, sex, occupation, education, satisfactory living condition, number of household contacts, number of household contacts under 18, whether currently in prison, whether homeless, whether receives monetary assistance, whether resides in urban area, whether reside outside Moldova more than 3 months in the past year, and whether resides in a district with low/median/high prevalence (<10%, 10–20%, and >20%) of resistance to FLQs.

2. TB-related information, including TB anatomic location (pulmonary, extrapulmonary), and TB type (e.g., new case, relapse case, previous treatment failure).

3. Results of microbiological tests, including microscopy.

Table 1 displays the distribution of values that these features take among patients with RR-TB, and patients with RR- and FLQ-resistant TB. To prepare our dataset, we coded entities with no or unrealistic values as "missing." We note that age, sex, and TB type did not have any missing values (Table 1). When training and evaluating our predictive models, we extrapolated the missing entries for two features 'number of household contacts' and 'number of household contacts 18 or younger' by replacing them with the mean values of each column. For all categorical features, as the occurrence of "missing" values may not be completely random, we kept all records with a "missing" value for these features when training and evaluating our predictive models.

The outcome we were interested in predicting was the resistance to FLQs, which we represented by a binary indicator. We defined resistance to FLQs as having a positive LJ or MGIT culture test result for at least one of FLQs (ofloxacin, levofloxacin, and/or moxifloxacin). Additional details about how resistance to different drugs was determined are provided in §S1.1 in S1 Text.

## Model development and evaluation

We followed the guidelines for the Transparent Reporting of a multivariable prediction model for Individual Prognosis Or Diagnosis (TRIPOD) to develop and evaluate the predictive models described here [46]. We considered three supervised machine learning models: logistic regression, neural network, and random forest [47] (see §S2 in S1 Text). For each model, we estimate the probability of resistance to FLQs given the characteristics of a patient and classify the patient as "infected with FLQ-resistant TB" if this estimated probability is greater than a preset (e.g., 0.5) classification threshold. To identify features with important predictive power and to remove features that would diminish the model's accuracy, we used three feature selection methods: recursive feature elimination [48], $L_1$ regularization [49], and permutation importance [50], all of which automate the selection of important features to optimize the model accuracy (§S2 in S1 Text).

To assess the internal validity of our models, we followed the bootstrap validation procedure recommended by The TRIPOD Statement (see §S4) to estimate the optimism-corrected area under the receiver operating characteristic curve (OC-AUC-ROC), and the optimism-corrected sensitivity, specificity, F1 score, and Matthews correlation coefficient (MCC) [46]. Compared to randomly splitting the dataset into model development and model validation sets, the bootstrap validation approach recommended by the TRIPOD Statement is shown to be a stronger approach as it utilizes the entire dataset for model development and validation

[51, 52]. Given that the relatively small size of our dataset ($N = 540$) did not allow for conducting temporal validation, we use the bootstrap validation approach recommended by the TRIPOD Statement to obtain estimates of the performance measures listed above.

In this context, sensitivity measures the probability that the model correctly detects FLQ-resistant TB, and specificity measures the probability that the model correctly detects susceptibility to FLQs. We compared the performances of models identified through different feature selection methods based on the estimated OC-AUC-ROC and selected the model with the highest estimated OC-AUC-ROC as the final model. To understand the importance of features, we recorded the number of times that each feature was identified as significant by the corresponding feature selection algorithm among each bootstrap iteration.

We note that the sensitivity and the specificity of predictive models considered here depend on the selected classification threshold (i.e., the probability above which we classify a patient as "infected with FLQ-resistant TB"). Selecting a lower classification threshold results in a more sensitive model at the expense of reduced specificity. Therefore, we evaluated our predictive models based on the estimates of OC-AUC-ROC and the estimates of sensitivity and specificity under varying classification thresholds.

## Impact on the selection of antituberculous medications

The most recent WHO-recommended standard RR-TB treatment regimens (both the shorter and longer options) include FLQs [9]. Therefore, if the prevalence of FLQ-resistant TB is $\mu$% among patients with RR-TB, following the WHO standardized regimen would result in $\mu$% of patients with RR-TB not receiving an appropriate treatment regimen (i.e., a regimen that is consistent with the susceptibility of their *M. tuberculosis* strain to FLQs). Following the hierarchy of the WHO grouping of anti-TB agents [9], we assume that when resistance to FLQs is suspected, the FLQ is replaced with delamanid (DLM). The use of predictive models to decide whether FLQs should be included or replaced with DLM, could improve the probability that a patient with RR-TB receives an appropriate treatment regimen (i.e., a treatment regimen that includes FLQs when susceptible to FLQs and that includes DLM in place of FLQs when resistant to FLQs). However, predictive models with low specificity would increase the unnecessary use of DLM, which consequently raises the risk for the selection of additional resistance to this drug.

To evaluate whether the use of the predictive models developed here could improve the selection of antibiotics for patients with RR-TB, we assessed the utility of each model using the net benefit measure [46], which is defined as $\lambda q(p) - c(p)$. Here, $p$ is the classification threshold, $q(p)$ is the expected proportion of individuals with RR-TB who receive an appropriate treatment regimen if the classification threshold is set to $p$, and $c(p)$ is the expected proportion of individuals with RR-TB who unnecessarily receive DLM (instead of FLQs) if the classification threshold is set to $p$ (see §S6 for additional details). In the above equation, $\lambda$ is a trade-off threshold that represents the policymaker's willingness to accept an increase in the proportion of individuals who unnecessarily receive DLM (i.e., $c(p)$) in order to increase the proportion of individuals who receive appropriate treatment regimens (i.e., $q(p)$). For example, $\lambda = 5$ implies that for every 1 percent point increase in the proportion of individuals who receive an appropriate treatment regimen, the policymaker is willing to accept 5 percent point increase in the proportion of individuals who unnecessarily receive DLM.

To examine how accounting for the local prevalence of FLQ-resistance would impact the performance of predictive models described above, we developed two classes of models, which differed based on whether they include or exclude the feature 'Residing in a region with low/medium/high prevalence of resistance to FLQs' (Table 1).

### Role of the funding source

The funders of the study had no role in study design, data collection, data analysis, data interpretation, or writing of the report. The corresponding author had full access to all the data in the study and all authors had final responsibility for the decision to submit for publication.

## Results

Based on the inclusion criteria displayed in Fig 1, 540 individuals with RR-TB diagnosed with Xpert MTB/RIF and confirmed positive or negative DST for FLQs were included to develop and evaluate the predictive models described above. Among these individuals, those with confirmed FLQ-resistant TB were more likely to live in regions with high prevalence of FLQ resistance and to have a history of treatment failure (Table 1).

All predictive models developed here resulted in OC-AUC-ROC estimates that were statistically greater than 0.5 (Table 2). Neural network models led to higher OC-AUC-ROC estimates compared with logistic regression and random forest models. Among models that did not account for the local prevalence of resistance to FLQs, the neural network model with permutation importance led to the highest OC-AUC-ROC estimate at 0.81 (0.77,0.85). Including the feature 'Residing in a region with low/medium/high prevalence of resistance to FLQs' (Table 1) to capture the local prevalence of resistance to FLQs increased the model's OC-AUC-ROC to 0.87 (0.83,0.91) (Table 2). Based on the estimates of OC-AUC-ROC, we chose the neural network model with features identified by permutation importance as our final model. The analyses presented below are based on this model.

Our final model included the following features: age, number of household contacts, number of household contacts 18 or younger, living in a district with low, medium, or high prevalence of FLQ-resistance, TB type (new or relapse), education level (secondary or primary), if unemployed, results of microscopy test, whether residing in an urban area, and whether the living condition is satisfactory. These features were selected as important in more than 50% of bootstrap iterations used to estimate OC-AUC-ROC of this model (Fig 2).

The current strategy of using the standardized regimen for all patients with RR-TB implicitly assumes the absence of resistance to FLQs, which can be regarded as a predictive model with 0% sensitivity (i.e., correctly identifying 0% of patients with FLQ-resistant TB) and 100% specificity (i.e., correctly identifying 100% of patients with a TB strain susceptible to FLQs). This corresponds to using the classification threshold 1 in our predictive models. Lowering this classification threshold increases the sensitivity of the model but decreases its specificity (Fig 3A). This, in consequence, increases the proportion of patients with RR-TB who receive appropriate treatment regimen that matches the susceptibility of their TB strains to FLQs but also increases the proportion of patients who may be unnecessarily treated with DLM (Fig 3B).

**Table 2. The estimated optimism-corrected area under the receiver operating characteristic curve (OC-AUC-ROC), for predictive models developed by different machine learning algorithms and feature selection methods.**

| Machine learning model | Logistic Regression | | | Neural Network | Random Forest | |
|---|---|---|---|---|---|---|
| Feature selection method | Recursive Feature Elimination | $L_1$ Regularization | Permutation Importance | Permutation Importance | Recursive Feature Elimination | Permutation Importance |
| Model without information on local prevalence of resistance to FLQs | 0.58 (0.52,0.63) | 0.57 (0.52,0.63) | 0.59 (0.53,0.65) | 0.81 (0.77,0.85) | 0.79 (0.74,0.83) | 0.61 (0.52,0.68) |
| Model with information on local prevalence of resistance to FLQs | 0.69 (0.63,0.73) | 0.68 (0.63,0.74) | 0.67 (0.61,0.72) | 0.87 (0.83,0.91) | 0.80 (0.76,0.83) | 0.76 (0.72,0.80) |

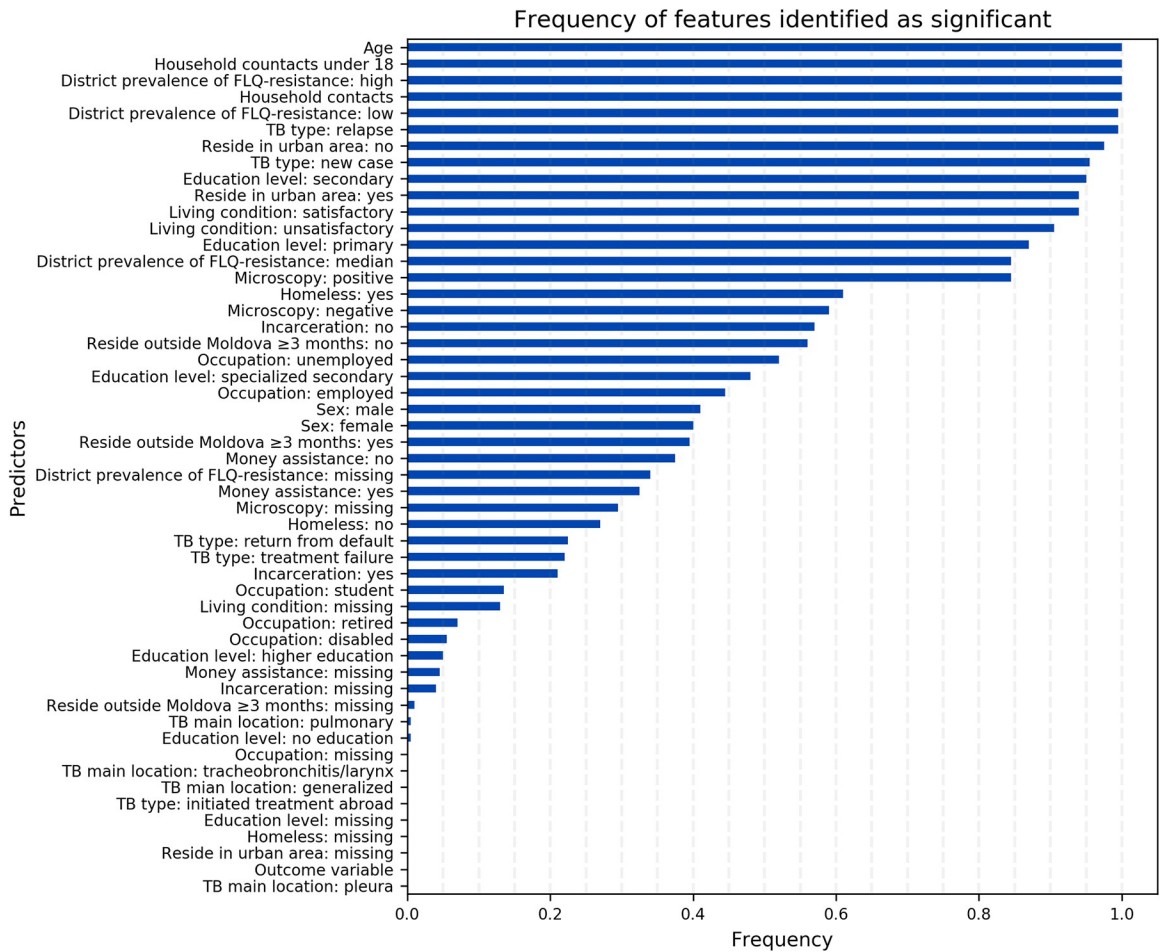

**Fig 2. The frequency of features identified as important through iterations of the bootstrap validation algorithm (§S4 in S1 Text) to evaluate the OC-AUC-ROC of our final model (neural network classifier and permutation importance algorithm).**

As the increase in the proportion of patients receiving appropriate treatment regimens is accompanied by an increase in the unnecessary use of DLM (Fig 3B), the optimal choice for the classification threshold depends on the policymaker's willingness to tradeoff between these two conflicting objectives. Fig 4 displays the classification thresholds that maximizes the net benefit of the neural network model for different values of trade-off threshold λ. At the trade-off threshold λ = 0, which represents the unwillingness to accept an increase in the unnecessary use of DLM even if that improves the proportion of patients receiving appropriate treatment regimens, the optimal classification threshold is 1. This results in a predictive model with 0% sensitivity and 100% specificity, which is equivalent to the current strategy of using the standardized treatment regimen for all patients with RR-TB. As the trade-off threshold λ increases, the optimal classification threshold reduces, resulting in predictive models with lower sensitivity and higher specificity. Our final neural network model had statistically higher net benefit than the current strategy of using the standardized treatment regimen for all patients with RR-TB for trade-off thresholds λ≥1.0 (Fig 4). Given the similar performance of neural network and random forest models in our study (Table 2), we also reported the performance of the random forest model with recursive feature elimination in §S7 in S1 Text.

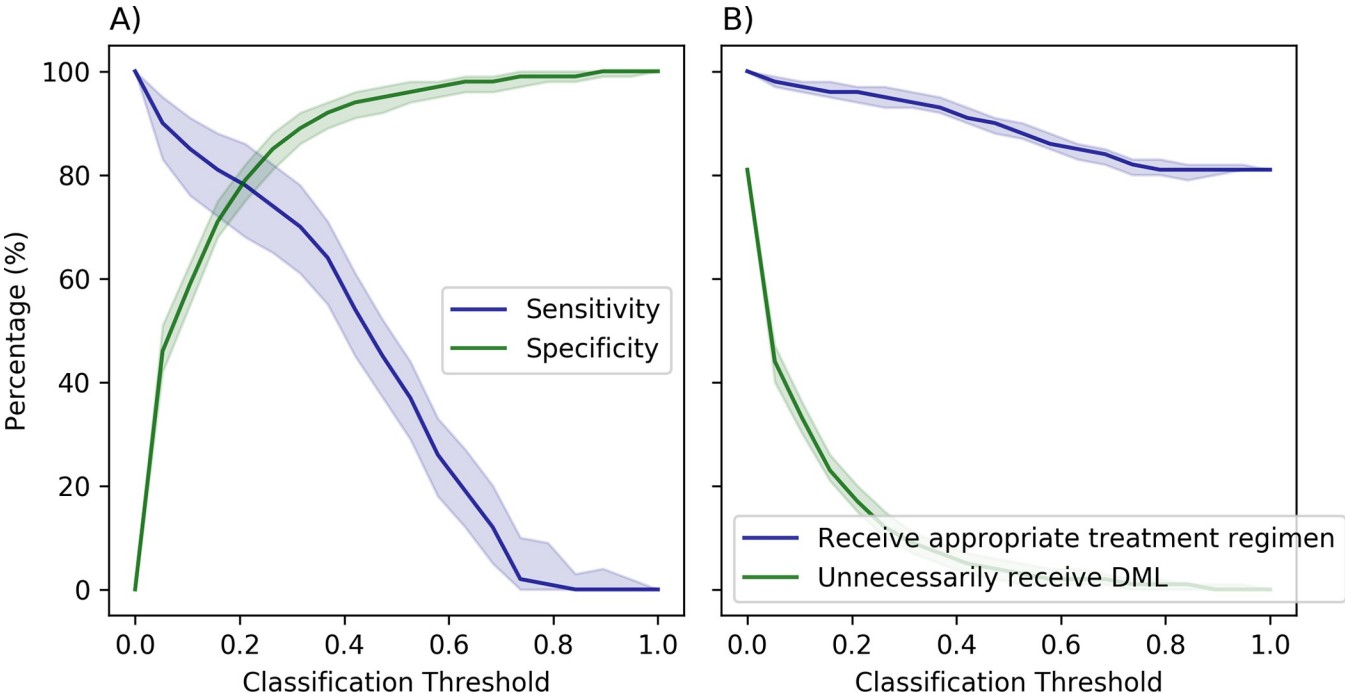

**Fig 3. Evaluating the performance of the neural network model that accounts for the local prevalence of resistance to FLQs using features identified by permutation importance for varying classification threshold.** The impact of the classification threshold on the optimism-corrected sensitivity and specificity is displayed in Panels A; the impact of the classification threshold on the optimism-corrected proportion of individuals receiving an appropriate treatment regimen (i.e., a regiment that is consistent with susceptibility of a patient's *M. tuberculosis* strain to FLQ) and on the optimism-corrected proportion of individuals who are unnecessarily treated with delamanid (DLM) is displayed in Panels B. The regions represent 95% bootstrap confidence intervals. See Figure F in S1 Text for estimates of F1 and Matthews correlation coefficient (MCC) scores for varying classification threshold.

## Discussion

Using data from the national TB surveillance system in the Republic of Moldova, we were able to develop predictive models to identify resistance to FLQs among patients diagnosed with RR-TB. Our final model, which was a neural network model with features identified by permutation importance method, had an OC-AUC-ROC of 0.87 (0.83,0.91). We identified age, number of household contacts, number of household contacts 18 or younger, living in districts with low, medium, or high prevalence of FLQ-resistance, TB type (new or relapse), education level (secondary or primary), if unemployed, results of microscopy test, whether residing in an urban area, and whether the living condition is satisfactory as strong features to indicate resistance to FLQs (Fig 2). These findings are consistent with the results of prior studies demonstrating that these characteristics were associated with the risk of resistance [28–34,37,38,41,42,43].

The sensitivity and the specificity of models developed here depend on the classification threshold used to predict the resistance to FLQs (Fig 3A and 3B). The optimal choice of this threshold requires a trade-off between the positive and negative consequences of using such predictive models with imperfect sensitivity and specificity. To demonstrate, we considered a scenario where FLQs can be replaced with DLM in the patient's treatment regimen if resistance to FLQs is suspected. Increasing this classification threshold would lower the proportion of patients who are unnecessarily treated with DLM (i.e., were falsely identified as FLQ-resistant) but would also decrease the proportion of patients who receive a treatment regimen that matches the susceptibility of their TB strains to FLQs (Fig 3C and 3D). Hence, the added utility

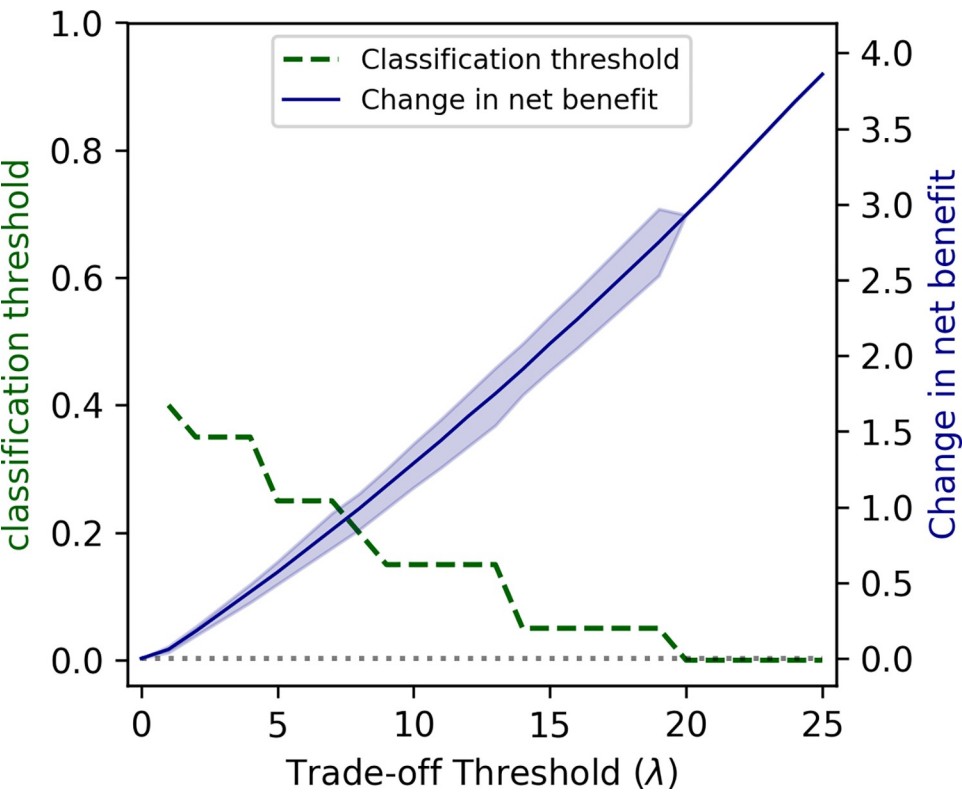

**Fig 4. The optimal choice of the classification threshold for varying values of the policymaker's trade-off threshold and the optimism-corrected utility of the neural network model to determine whether FLQs should be included or replaced with DLM for a patient with RR-TB.** The model's utility is measured as the change in net benefit with respect to the strategy that uses the standardized treatment regimens for all patients with RR-TB. The trade-off threshold $\lambda$ represents the percentage point increase in the proportion of individuals unnecessarily treated with DLM that the policymaker is willing to tolerate to increase the proportion of individuals who receive appropriate treatment by 1 percentage point. The regions represent 95% bootstrap confidence intervals.

of our predictive models depends on the policymaker's trade-off threshold $\lambda$ that represents their willingness to increase the proportion of individuals who unnecessarily receive DLM in order to increase the proportion of individuals who receive appropriate treatment regimens. Our analysis suggests that, compared to the strategy of using the standardized regimen for all patients with RR-TB, the added utility of our predictive models is statistically significant for a policymaker who is willing to accept $\geq 1.0$ percent point increase in the proportion of individuals who unnecessarily receive DLM for every 1 percent point increase in the proportion of individuals who receive appropriate treatment regimens.

We acknowledge several limitations in this study. First, our models are based only on data from the Republic of Moldova and were not externally validated on datasets from other settings. Therefore, the generalizability of these models to other settings remains unknown. Second, our predictive models were developed using a relatively small number of observations ($n = 540$) and hence, the performance of these models may be sensitive to the training data. By using optimism-corrected estimates for performance metrics to evaluate the models developed here, we believe the uncertainties due to the small sample size is properly accounted for in our conclusions [46]. Third, we used a simple approach to incorporate the local information about the prevalence of resistance to FLQs. The information about the resistance to FLQs was available for only 671 individuals with RR-TB and therefore, we were not able to estimate the

prevalence of resistance to FLQs for all 50 districts of Moldova. Instead, we used a single categorical feature to capture whether an individual resides in a region with low, median, or high prevalence (<10%, 10–20%, and >20%) of resistance to FLQs.

We measured the utility of a predictive model based its impact on the proportion of patients with RR-TB who would receive an appropriate treatment regimen or would be unnecessarily treated with DLM. The utility of predictive models should be ideally investigated using a cost-effectiveness analysis that quantifies the cost and health consequences of replacing FLQs with DLM, which could reduce the clinical longevity of DLM but improve the treatment outcomes of patients with RR-TB. Accounting for this tradeoff allows the decisionmaker to identify the optimal classification threshold based on their tolerance in accepting a shorter clinical longevity for DLM if that leads to improving the treatment outcomes of patients with RR-TB.

To measure the impact of the proposed predictive models on the selection of antituberculous medications, we considered a simple scenario where FLQs could be replaced with DLM in the empiric treatment of RR-TB if resistance to FLQs is suspected. Under certain operational research conditions, new regimens (e.g., the Nix-TB regimen [53,54]) could be recommended for people with FLQ-resistant TB. The predictive models described here could also be useful under these scenarios to identify, at the point-of-care, individuals with RR-TB who would be eligible for these novel regimens.

In the future, rapid point-of-care DSTs are expected to become available [2,55]. For example, sputum-based Xpert MTB/XDR has been developed for rapid detection of resistance to isoniazid, fluoroquinolone, and ethionamide [56]. While rapid point-of-care DSTs could mitigate the limitations of current DSTs, they are not expected to be widely available in many high-burden settings. Moreover, these molecular-based DSTs can only identify the known genetic determinants of resistance. Therefore, the sensitivity of these tests is impacted by the prevalence of *M. tuberculosis* with gene mutations that are not associated with resistance. As such, phenotypic surveillance systems will continue to be maintained to ensure the sensitivity of these molecular-based tests. Therefore, even if rapid point-of-care DSTs become widely available, the predictive models described here could improve the accuracy of diagnosis by considering data from phenotypic surveillance systems of drug-resistant TB and including the results of rapid DSTs or whole genome sequencing as features.

In the absence of rapid point-of-care DSTs, the treatment of individuals with RR-TB in many high-burden settings, remains empiric and according to standardized regimens. Although standardized regimens facilitate access to second-line treatment in high-burden settings, they may lead to poor treatment outcomes, unnecessary toxicity, increased risk for emergence of additional resistance, and further transmission of drug-resistant TB [1–4,12,18–27,57] To improve the selection of antibiotics for patients with RR-TB, we showed that data from national phenotypic surveillance systems of drug-resistant TB could be used to identify resistance to second-line anti-TB agents based on the patient's demographic and clinical information and the estimate of the local prevalence of drug-resistant TB. Future studies could investigate the potential of these predictive models to optimize the selection of antibiotics, at the point-of-care, for patients with drug-resistant TB.

## Supporting information

**S1 Text. Additional Details and Analyses.** Table A. The estimated area under the receiver operating characteristic curve (AUC-ROC) calculated using 5-fold cross-validation, for predictive models developed by different machine learning algorithms and feature selection methods. Figure A. The estimated area under the ROC curves (AUC-ROC) of predictive models that did not account for the local prevalence of resistance to FLQs, identified by different feature

selection methods. Figure B. The estimated area under the ROC curves (AUC-ROC) of predictive models that accounted for the local prevalence of resistance to FLQs, identified by different feature selection methods. Figure C. The frequency of features identified as important using recursive feature elimination algorithm and random forest classifier among 200 bootstrap iterations (see §S4 in S1 Text). Figure D. Evaluating the performance of the random forest model that accounts for the local prevalence of resistance to FLQs using features identified by recursive feature elimination for varying classification threshold. Figure E. The optimal choice of the classification threshold for varying values of the policymaker's trade-off threshold and the optimism-corrected utility of the random forest model to determine whether FLQs should be included or replaced with DLM for a patient with RR-TB. Figure F. The estimates F1 and Matthews correlation coefficient (MCC) scores for varying classification threshold for our final model (neural network classifier and permutation importance algorithm).
(PDF)

**S1 Dataset. Records of patients diagnosed with tuberculosis in Moldova between January 2018 and December 2019.**
(XLSX)

## Author Contributions

**Conceptualization:** Ted Cohen, Reza Yaesoubi.

**Data curation:** Shiying You, Melanie H. Chitwood, Kenneth S. Gunasekera, Valeriu Crudu.

**Formal analysis:** Shiying You.

**Funding acquisition:** Ted Cohen, Reza Yaesoubi.

**Investigation:** Shiying You, Kenneth S. Gunasekera, Reza Yaesoubi.

**Methodology:** Shiying You, Reza Yaesoubi.

**Software:** Shiying You.

**Supervision:** Jennifer Furin, Ted Cohen, Joshua L. Warren, Reza Yaesoubi.

**Validation:** Shiying You.

**Visualization:** Shiying You.

**Writing – original draft:** Shiying You, Reza Yaesoubi.

**Writing – review & editing:** Shiying You, Melanie H. Chitwood, Kenneth S. Gunasekera, Valeriu Crudu, Alexandru Codreanu, Nelly Ciobanu, Jennifer Furin, Ted Cohen, Joshua L. Warren, Reza Yaesoubi.

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
