## [Decision Letter · Decision Letter 0]

12 Dec 2021

PDIG-D-21-00020

Predicting resistance to fluoroquinolones among patients with rifampicin-resistant tuberculosis using machine learning methods

PLOS Digital Health

Dear Dr. Yaesoubi,

Thank you for submitting your manuscript to PLOS Digital Health. After careful consideration, we feel that it has merit but does not fully meet PLOS Digital Health’s publication criteria as it currently stands. Therefore, we invite you to submit a revised version of the manuscript that addresses the points raised during the review process.

In particular, the reviewers raise a number of important methodological questions, concerns, and recommendations that should be carefully addressed by the authors (both via a point-by-point response letter and in the revised manuscript text) prior to further consideration.

We look forward to receiving your revised manuscript.

Kind regards,

Maimuna Majumder, PhD

Academic Editor

PLOS Digital Health

Journal Requirements:

1. We ask that a manuscript source file is provided at Revision. Please upload your manuscript file as a .doc, .docx, .rtf or .tex. If you are providing a .tex file, please upload it under the item type ‘LaTeX Source File’ and leave your .pdf version as the item type ‘Manuscript’.

2. Please provide separate figure files in .tif or .eps format only and remove any figures embedded in your manuscript file. Please ensure that all files are under our size limit of 20MB.  

For more information about how to convert your figure files please see our guidelines: https://journals.plos.org/digitalhealth/s/figures

Once you've converted your files to .tif or .eps, please also make sure that your figures meet our format requirements.

3. We notice that your supplementary figures amd tables are included in the manuscript file. Please remove them and upload them with the file type 'Supporting Information'. Please ensure that all Supporting Information files are included correctly and that each one has a legend listed in the manuscript after the references list. 

4. Please provide us with a direct link to the base layer of the map used in Figure S1 and ensure this location is also included in the figure legend. 

Please note that, because all PLOS articles are published under a CC BY license (creativecommons.org/licenses/by/4.0/), we cannot publish proprietary maps such as Google Maps, Mapquest or other copyrighted maps. If your map was obtained from a copyrighted source please amend the figure so that the base map used is from an openly available source.

Please note that only the following CC BY licences are compatible with PLOS licence: CC BY 4.0, CC BY 2.0  and CC BY 3.0, meanwhile such licences as CC BY-ND 3.0 and others are not compatible due to additional restrictions. If you are unsure whether you can use a map or not, please do reach out and we will be able to help you. 

The following websites are good examples of where you can source open access or public domain maps:

5. Please provide additional details regarding participant consent. In the ethics statement in the Methods section, please ensure that you have specified (a) whether consent was informed and (b) what type you obtained (for instance, written or verbal, and if verbal, how it was documented and witnessed). If your study included minors, state whether you obtained consent from parents or guardians. If the need for consent was waived by the ethics committee, please include this information.

Reviewers' comments:

Reviewer's Responses to Questions

**Comments to the Author**

1. Does this manuscript meet PLOS Digital Health’s publication criteria? Is the manuscript technically sound, and do the data support the conclusions? The manuscript must describe methodologically and ethically rigorous research with conclusions that are appropriately drawn based on the data presented.

Reviewer #1: No

Reviewer #2: Yes

Reviewer #3: Partly

2. Has the statistical analysis been performed appropriately and rigorously?

Reviewer #1: N/A

Reviewer #2: Yes

Reviewer #3: Yes

3. Have the authors made all data underlying the findings in their manuscript fully available (please refer to the Data Availability Statement at the start of the manuscript PDF file)?

Reviewer #1: Yes

Reviewer #2: Yes

Reviewer #3: Yes

4. Is the manuscript presented in an intelligible fashion and written in standard English?

Reviewer #1: Yes

Reviewer #2: Yes

Reviewer #3: Yes

5. Review Comments to the Author

Reviewer #1: This work used single site RR-MTB data to build machine learning models. The input includes heterogeneous features collected for national surveillance, e.g. demographics, TB testing, etc. A trade-off strategy is proposed for determining classification threshold. Several comments are as follows,

1. P6, the equation for determining classification threshold. If FLQ-resistance is positive, the q(p) denotes true negative (FLQ-susceptible case that receives the treatment regime with FLQ properly) and c(p) is false positive (FLQ-susceptible case that receives DLM unnecessarily). Contrarily, it is stated in P8 that the threshold is determined through trade-off between false positive and false negative. Please clarify.

2. The most important details of experiments are missing, e.g., data split, the parameter setting of different machine learning models and feature selection scheme. Therefore, it’s difficult to answer the following questions: (1) which model is the best? (2) how are the reported results obtained?

3. What are the selected features that generate the best performance?

4. The most important pre-processing details are missing. How are the categorical input features converted to numerical input?

Reviewer #2: We have read “Predicting resistance to fluoroquinolones among patients with rifampicin-resistant tuberculosis using machine learning methods” by You et al. The authors developed a machine-learning predictor for fluoroquinolone resistance of rifampicin resistance tuberculosis (TB) based on patient demographics, TB clinical information and disease prevalence features. Predictions from this model could be used to select antibiotics more rationally for treatment of TB in the future. We appreciated the author’s clear description of the methods, results and implications as well as their honest discussion of limitations. Overall, we only have minor questions regarding feature interpretation, model training and the necessity of machine learning for this application given the relative few number of features.

1. We were excited to see that the models were trained using demographic and surveillance information (rather than genomic data) because it could reasonably be rolled out quickly without new equipment. However, we were disappointed that relatively little time was spent discussing the most important features to model predictions. We feel the manuscript would benefit from more interpretation of important features leading to model predictions (e.g. was homelessness a major factor? What about TB prevalence? We are familiar with some nice ways to evaluate feature importance for random forests, e.g. permutation, MDA, etc. A description of feature importance for the best models would be a valuable addition to this manuscript.

2. The authors don’t discuss model cross validation in-depth but based on the methods section it looks like they did some cross validation with TRIPOD recommended bootstrapping. Have the authors compared this method with others like leave-one-out or KFolds? We understand that the sample size is relatively smally and class sizes are unbalanced but it would still be nice to understand how model predictions fare with a larger hold-out group. This would also inform how potentially generalizable such a model might be to other regions.

3. The authors don’t discuss the results from their neural networks much. We are interested in their opinion of why the neural networks did not perform as well as other methods. Why did they opt to remove prevalence of FLQ from this model in particular?

Reviewer #3: The manuscript: “Predicting resistance to fluoroquinolones among patients with rifampicin-resistant tuberculosis using machine learning methods” describes an effort to detect resistance to second line fluoroquinolones by using data from rifampicin-resistant TB samples as delineated by GeneXpert-MTB/RIF, in samples present in Moldova. The authors use the optimism corrected AUROC metric to justify the performance of their models, which were built using logistic regression, neural network and random forest algorithms.

Below are my general comments per section.

Methods: Data source and study population

It is unclear in the Methods section as to why and how there are cases in Table one which have been diagnosed with RR-TB through Xpert-MTB/RIF, but were negative for rifampicin resistance, presumably using other DST methods.

Methods: Predictors and outcomes

Overall, the idea behind using Expert-MTB/RIF data to detect resistance to FLQs is innovative, however, I am concerned about the general interpretability of the model considering the range of features used – particularly as certain features may be correlated to disease prevalence, but not necessarily contribute to FLQ resistance. Further to that, it is unclear how the authors account for missing data in cases where only partial demographic, TB-related or microbiological tests are present. Supplementary section 1.2 briefly brushes on this point – however the authors do not explicitly refer to this in the main text, while it is unclear whether supervised or semi-supervised machine learning was adopted to develop the models, considering missing data.

Methods: Model development and evaluation

The readers would benefit from an explanation of the differences between the three feature selection methods adopted. Further to that, it would also be useful for the readers to understand why the TRIPOD recommendations were used, rather than separating the data into train/test and using metrics other than OC-AUROC to assess model performance. This is especially because training and testing on the same data offers no generalizability potential for the model (as the readers highlight), meaning that its real-world clinical utility cannot be quantified. It would perhaps be more useful if both types of models were presented, and final features also compared. Additionally, reporting only OC-AUROC for the TRIPOD-based model is not enough to assess performance, and more balanced features, like MCC or F1 score may prove additional information.

Results

While a list of selected features has been made available by the authors in the supplementary files, and highlighted in the discussion, the authors do not investigate the importance and relevance of these features further, which would be especially informative to interpret the model itself, and its potential utility in the clinic.

It is not especially clear how the authors investigated the replacement of FLQs with DLM – was it through mathematical modelling or machine learning? Also, is there a possible reason why resistance to one FLQ does not warrant switching to another FLQ before using the last-line drug DLM? This needs to be made clear within the main text.

Discussion

While suggesting the replacement of FLQs with DLM and the different thresholds applied, it is important for the readers to understand the implications of using DLM haphazardly in the clinic – meaning that stats pertaining to current TB resistance (if any) to DLM need to be highlighted, as well as the importance of DLM stewardship to increase its clinical longevity. This includes a discussion on what a >= 2.0 percent point increase would mean if policymakers chose to take the risk.

6. PLOS authors have the option to publish the peer review history of their article (what does this mean?). If published, this will include your full peer review and any attached files.

**Do you want your identity to be public for this peer review?** For information about this choice, including consent withdrawal, please see our Privacy Policy.

Reviewer #1: **Yes: **YANG YANG

Reviewer #2: No

Reviewer #3: No

---

## [Editor Report · Decision Letter 1]

9 May 2022

Predicting resistance to fluoroquinolones among patients with rifampicin-resistant tuberculosis using machine learning methods

PDIG-D-21-00020R1

Dear Dr. Yaesoubi,

We are pleased to inform you that your manuscript 'Predicting resistance to fluoroquinolones among patients with rifampicin-resistant tuberculosis using machine learning methods' has been provisionally accepted for publication in PLOS Digital Health.

Best regards,

Maimuna Majumder, PhD

Academic Editor

PLOS Digital Health